# Does the human placenta express the canonical cell entry mediators for SARS-CoV-2?

Roger Pique-Regi[1,2,3]*, Roberto Romero[1,2,4,5,6,7]*, Adi L Tarca[1,3,8], Francesca Luca[2,3], Yi Xu[1,3], Adnan Alazizi[2], Yaozhu Leng[1,3], Chaur-Dong Hsu[1,3,9], Nardhy Gomez-Lopez[1,3,10]*

[1]Perinatology Research Branch, Division of Obstetrics and Maternal-Fetal Medicine, Division of Intramural Research, *Eunice Kennedy Shriver* National Institute of Child Health and Human Development, National Institutes of Health, U.S. Department of Health and Human Services, Detroit, United States; [2]Center for Molecular Medicine and Genetics, Wayne State University School of Medicine, Detroit, United States; [3]Department of Obstetrics and Gynecology, Wayne State University School of Medicine, Detroit, United States; [4]Department of Obstetrics and Gynecology, University of Michigan, Ann Arbor, United States; [5]Department of Epidemiology and Biostatistics, Michigan State University, East Lansing, United States; [6]Detroit Medical Center, Detroit, United States; [7]Department of Obstetrics and Gynecology, Florida International University, Miami, United States; [8]Department of Computer Science, Wayne State University College of Engineering, Detroit, United States; [9]Department of Physiology, Wayne State University School of Medicine, Detroit, United States; [10]Department of Biochemistry, Microbiology and Immunology, Wayne State University School of Medicine, Detroit, United States

**\*For correspondence:**
rpique@wayne.edu (RP-R);
prbchiefstaff@med.wayne.edu
(RR);
ngomezlo@med.wayne.edu (NG-L)

**Competing interests:** The authors declare that no competing interests exist.

**Abstract** The pandemic of coronavirus disease 2019 (COVID-19) caused by the severe acute respiratory syndrome coronavirus 2 (SARS-CoV-2) has affected more than 10 million people, including pregnant women. To date, no consistent evidence for the vertical transmission of SARS-CoV-2 exists. The novel coronavirus canonically utilizes the angiotensin-converting enzyme 2 (ACE2) receptor and the serine protease TMPRSS2 for cell entry. Herein, building upon our previous single-cell study (Pique-Regi et al., 2019), another study, and new single-cell/nuclei RNA-sequencing data, we investigated the expression of ACE2 and TMPRSS2 throughout pregnancy in the placenta as well as in third-trimester chorioamniotic membranes. We report that co-transcription of ACE2 and TMPRSS2 is negligible in the placenta, thus not a likely path of vertical transmission for SARS-CoV-2. By contrast, receptors for Zika virus and cytomegalovirus, which cause congenital infections, are highly expressed by placental cell types. These data show that the placenta minimally expresses the canonical cell-entry mediators for SARS-CoV-2.

## Introduction

The placenta serves as the lungs, gut, kidneys, and liver of the fetus (*Burton and Jauniaux, 2015*; *Maltepe and Fisher, 2015*). This fetal organ also has major biological actions that modulate maternal physiology (*Burton and Jauniaux, 2015*; *Sasaki and Norwitz, 2011*; *Taglauer et al., 2014*; *Fitzgerald et al., 2018*) and, importantly, together with the extraplacental chorioamniotic membranes, shield the fetus against microbes from hematogenous dissemination and from invading the amniotic cavity (*Ander et al., 2019*; *Kwon et al., 2014*). Indeed, most pathogens that cause

hematogenous infections in the mother cannot reach the fetus, which is largely due to the potent protective mechanisms provided by placental cells (i.e. trophoblast cells: syncytiotrophoblasts and cytotrophoblasts) (*Parry et al., 1997*; *Koi et al., 2002*; *Arora et al., 2017*). Yet, some of these pathogens such as *Toxoplasma gondii*, Rubella virus, herpesvirus (HSV), cytomegalovirus (CMV), and Zika virus (ZIKV), among others, are capable of crossing the placenta and infecting the fetus, thus causing congenital disease (*Stegmann and Carey, 2002*; *Coyne and Lazear, 2016*).

In December 2019, a local outbreak of pneumonia caused by a novel coronavirus—severe acute respiratory syndrome coronavirus 2 (SARS-CoV-2)—was reported in Wuhan (Hubei, China) (*Dong et al., 2020a*). After exposure to SARS-COV-2, susceptible individuals can develop coronavirus disease 2019 (COVID-19) consisting of symptoms that may range from fever and cough to severe respiratory illness; in some cases, COVID-19 is life-threatening (*Centers for Disease Control and Prevention, 2020a*; *Wadman, 2020*). Since the onset of the outbreak, more than 10 million COVID-19 cases have been confirmed, accounting for more than 500,000 deaths (*COVID-19, 2020*). This pandemic has now spread throughout the entire world with recent epicenters in Europe (Italy and Spain) and the United States. By April 2019, the states of New York and Michigan were the most severely affected (*COVID-19, 2020*), given that the metropolitan areas of New York City and Detroit possess large populations subject to health disparities that include limited access to health care, chronic exposure to pollution, and pre-existing cardiovascular conditions (*Centers for Disease Control and Prevention, 2020b*).

Pregnant women and their fetuses represent a potential high-risk population in light of the COVID-19 outbreak (*Dashraath et al., 2020*; *Liu et al., 2020*; *Rasmussen et al., 2020*; *Ashokka et al., 2020*; *Della Gatta et al., 2020*; *Weber LeBrun et al., 2020*; *Tekbali, 2020*; *Vintzileos, 2020*; *Lokken, 2020*) given that viral infections such as influenza (*Neuzil et al., 1998*; *Lindsay et al., 2006*; *Jamieson et al., 2006*; *Cervantes-Gonzalez and Launay, 2010*; *Pandemic H1N1 Influenza in Pregnancy Working Group et al., 2010*; *Mosby et al., 2011*; *Pazos et al., 2012*; *Raj et al., 2014*), varicella (*Triebwasser et al., 1967*; *Paryani and Arvin, 1986*; *Esmonde et al., 1989*; *Haake et al., 1990*; *Swamy and Dotters-Katz, 2019*), Ebola (*Olgun, 2018*; *Muehlenbachs et al., 2017*), and measles (*Christensen et al., 1954*; *Atmar et al., 1992*) show increased severity in this physiological state. Other coronaviruses, such as SARS-CoV-1 and MERS-CoV, have severe effects in both the mother and the fetus, but vertical transmission has not been proven (*Wong et al., 2004*; *Ng et al., 2006*; *Alserehi et al., 2016*; *Jeong et al., 2017*), albeit these studies included very few cases. By contrast with the above-mentioned viral infections, only ~15% of pregnant women test positive for SARS-CoV-2 and a small fraction of them are symptomatic (*Sutton et al., 2020*), most of whom experience only a mild illness (*Chen et al., 2020a*). Consequently, the clinical characteristics of pregnant women with COVID-19 appear similar to those of non-pregnant adults (*Yu et al., 2020*). Yet, recent reports have shown that severe COVID-19 can lead to maternal death (*Hantoushzadeh et al., 2020*; *Blitz, 2020*). However, thus far, no conclusive evidence of vertical transmission has been generated (*Weber LeBrun et al., 2020*; *Chen et al., 2020b*; *Stower, 2020*; *Lamouroux et al., 2020*; *Yan et al., 2020*; *Siberry et al., 2020*). Consistently, infants born to mothers with COVID-19 test negative for SARS-CoV-2, do not develop serious clinical symptoms (e.g., fever, cough, diarrhea, or abnormal radiologic or hematologic evidence), and are promptly discharged from the hospital (*Chen et al., 2020c*). Nevertheless, new evidence has emerged suggesting that the fetus can respond to SARS-CoV-2 infection.

Case reports have shown that a small fraction of neonates born to women with COVID-19 tested positive for the virus at 1–4 days of life (*Zeng et al., 2020a*; *Alzamora et al., 2020*), yet these neonates subsequently tested negative on day 6-7 (*Zeng et al., 2020a*). In addition, serological studies revealed that a few neonates born to mothers with COVID-19 had increased concentrations of SARS-CoV-2 immunoglobulin (Ig)M as well as IgG (*Zeng et al., 2020b*; *Dong et al., 2020b*). The elevated concentrations of IgG are likely due to the passive transfer of this immunoglobulin from the mother to the fetus across the placenta. However, the increased levels of IgM suggest that the fetus was infected with SARS-CoV-2 given that this immunoglobulin cannot cross the placenta as a result of its large molecular weight. Nonetheless, all neonates included in the above-mentioned studies

tested negative for the virus and did not present any symptoms (*Zeng et al., 2020b*; *Dong et al., 2020b*).

More recently, two case reports indicated that SARS-CoV-2 RNA has been detected in the amniotic fluid and placental tissues. In the first case report, the viral RNA was detected in amniotic fluid from a woman who was severely affected and died of COVID-19 (*Zamaniyan, 2020*). The premature neonate tested negative for SARS-CoV-2 after delivery but 24 hr later tested positive (*Zamaniyan, 2020*). In the second case report, the viral RNA was detected in the placenta and umbilical cord from a woman with severe pre-eclampsia, placental abruption, and other complications, yet none of the fetal tissues tested positive (*Hosier, 2020*). Therefore, whether SARS-CoV-2 can reach the fetus by crossing the placenta is still unclear.

## Results and discussion

Cell entry and the spread of SARS-CoV-2 are widely thought to depend on the angiotensin-converting enzyme 2 (ACE2) receptor (*Shang et al., 2020*; *Wang et al., 2020a*) and the serine protease TMPRSS2 (*Hoffmann et al., 2020*). In the study herein, we investigated whether the receptors responsible for SARS-CoV-2 infection are expressed in the human placenta (including the decidual tissues) throughout the three trimesters of pregnancy by using publicly available single-cell RNA-sequencing (scRNA-seq) data (*Vento-Tormo et al., 2018*; *Pique-Regi et al., 2019*) together with newly generated data (*Supplementary file 1*).

Strikingly, we found that very few cells co-express ACE2 and TMPRSS2 (*Figure 1A and B*). Using a very permissive threshold of expression of one transcript per cell, only four cells with co-expression

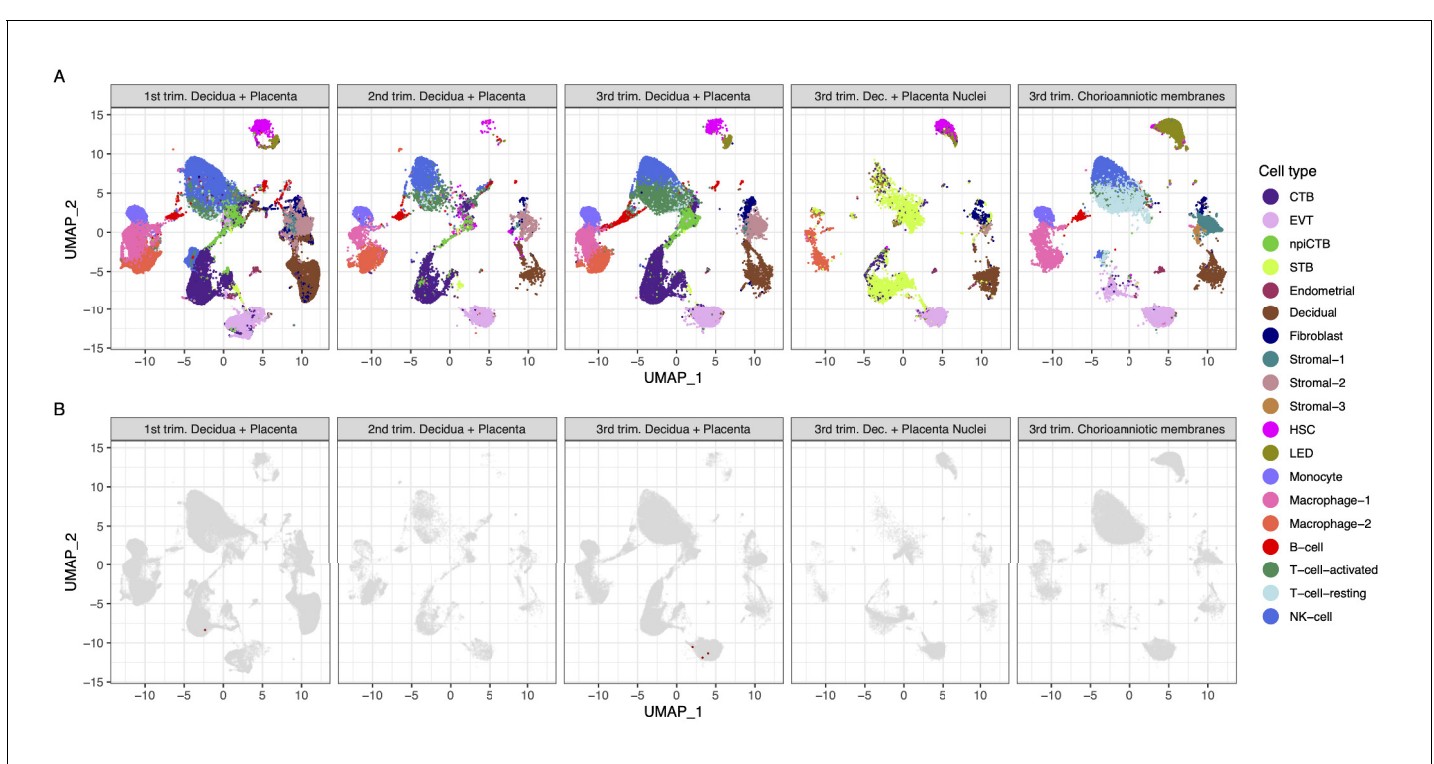

**Figure 1.** Transcriptional map of the human placenta, including the decidua, in the three trimesters of pregnancy. (**A**) Uniform Manifold Approximation Plot (UMAP), where dots represent single cells/nuclei and are colored by cell type (abbreviations used are: STB, Syncytiotrophoblast; EVT, Extravillous trophoblast; CTB, cytotrophoblast; HSC, hematopoietic stem cell; npiCTB, non-proliferative interstitial cytotrophoblast; LED, lymphoid endothelial decidual cell) (**B**) UMAP plot with cells/nuclei co-expressing one or more transcripts for ACE2 and TMPRSS2, genes that are necessary for SARS-CoV-2 viral infection and spreading, in red.

The online version of this article includes the following figure supplement(s) for figure 1:

**Figure supplement 1.** UMAP highlighting cells expressing syncytiotrophoblast genes CGA + CSH1 + GH2 that are more efficiently captured when isolating nuclei in snRNA-seq.

were detected in any of the three trimesters, resulting in an estimated <1/10,000 cells. Our first-trimester data are in agreement with a prior report showing minimal expression of ACE2 at the human maternal-fetal interface (*Zheng et al., 2020*); however, the same dataset was recently used to report the opposite (*Li et al., 2020*). Nonetheless, the co-expression of ACE2 and TMPRSS2 was not examined by either study, and it is important to consider log-transformation and data analysis issues for low-expressed genes, focusing on the fraction of cells expressing the transcripts (*Booeshaghi and Pachter, 2020*). We also evaluated the expression of SARS-CoV-2 receptors in the chorioamniotic membranes (also known as the extraplacental membranes) in the third trimester; these tissues may also serve as a point of entry for microbial invasion of the amniotic cavity and potentially the fetus (*Kim et al., 2015*). Again, co-expression of ACE2 and TMPRSS2 was minimally detected in the chorioamniotic membranes (*Figure 1A and B*).

A challenge in scRNA-seq studies is generating high-quality, single-cell suspensions containing both rare and difficult-to-dissociate (e.g., multinucleated cells) cell types. This is likely the reason why the reported scRNA-seq studies of the human placenta contain a low fraction of syncytiotrophoblast cells [STB, multinucleated cells forming the outermost fetal component of the placenta in direct contact with the maternal circulation (i.e., intervillous space)] (*Vento-Tormo et al., 2018*; *Pique-Regi et al., 2019*; *Tsang et al., 2017*). Therefore, we considered whether the expression of ACE2 and TMPRSS2 was minimally observed in the placental cell types as a result of the reduced fraction of STB cells (i.e., dissociation bias). To address this possibility, we prepared single-nucleus suspensions of the placental tissues (including the decidua basalis) and performed single-nuclear RNAseq (snRNA-seq), which reduces the dissociation bias against large cells (*Wu et al., 2019*). An important advantage of snRNA-seq is its compatibility with biobank-frozen samples; therefore, we pooled 32 placental villi/decidua samples collected in the third trimester (*Supplementary file 2*). This represents the first snRNA-seq study of the placental tissues. As expected, a larger fraction of STB cells/nuclei was observed using snRNA-seq compared to scRNA-seq (*Figure 1A* and *Figure 1—figure supplement 1*). Consistent with the scRNAseq analyses, the snRNAseq data demonstrated that co-expression of ACE2 and TMPRSS2 is unlikely in the placental tissues (*Figure 1B*). A limitation of snRNA-seq is that it has a higher background compared to scRNAseq and could capture ACE2 and TMPRSS2 transcripts from other cell types, but this should not affect the analyses reported herein because co-expression is not observed.

Finally, we explored the expression of ACE2 and TMPRSS2 in third-trimester placental tissues by mining two microarray datasets that we have previously reported (*Kim et al., 2009*; *Toft et al., 2008*). These analyses of bulk gene expression data revealed that, while ACE2 was detected above background in most of the samples, TMPRSS2 was largely undetected (*Supplementary file 3*). Collectively, these results consistently indicate that the human placental tissues negligibly co-express ACE2 and TMPRSS2. This reduced expression contrasts with the high expression of ACE2 and TMPRSS2 in nasal goblet and ciliated cells within the human airways, lungs, and gastrointestinal tract, which are targeted during COVID-19 (*Muus, 2020*; *Wang, 2020b*; *HCA Lung Biological Network et al., 2020*). Therefore, our results suggest that vertical transmission of SARS-CoV-2 is unlikely to occur unless facilitated by other concomitant pathological conditions resulting in a breach of the maternal-fetal crosstalk.

There is a possibility, however, that SARS-CoV-2 could infect the human placenta by using alternate entry routes while interacting with other proteins (*Gordon et al., 2020*). The expression of additional SARS-CoV-2-related receptors or proteins in the human placenta is shown in *Figure 2*, CoV-Alt; however, further research is required to test their participation in the pathogenesis of COVID-19. For example, in vitro studies suggest that BSG (Basigin, also called CD147 or EMMPRIN, transmembrane glycoprotein belonging to the immunoglobulin superfamily) provides an alternate entry for SARS-CoV-2 when ACE2 and TMPRSS2 are not expressed (*Blanco-Melo et al., 2020*; *Wang, 2020c*; *Ulrich and Pillat, 2020*). We found that the placenta and chorioamniotic membranes expressed high levels of BSG throughout pregnancy (*Figure 2*, CoV-Alt), yet this transcript is also widely expressed in all human tissues and cell types (*Figure 2—figure supplement 1*). Therefore, it is unlikely that this protein alone is a sufficient requirement for SARS-CoV-2 viral entry, and other proteins may be required to explain the cell type primarily affected by COVID-19. Moreover, cathepsin L (CSTL) and FURIN may also function as proteases priming the SARS-CoV-2 S protein (*Lukassen et al., 2020*). We found that these proteases are highly expressed by the placental tissues throughout gestation (*Figure 2*, CoV-Alt). Nevertheless, these proteases may not provide sufficient

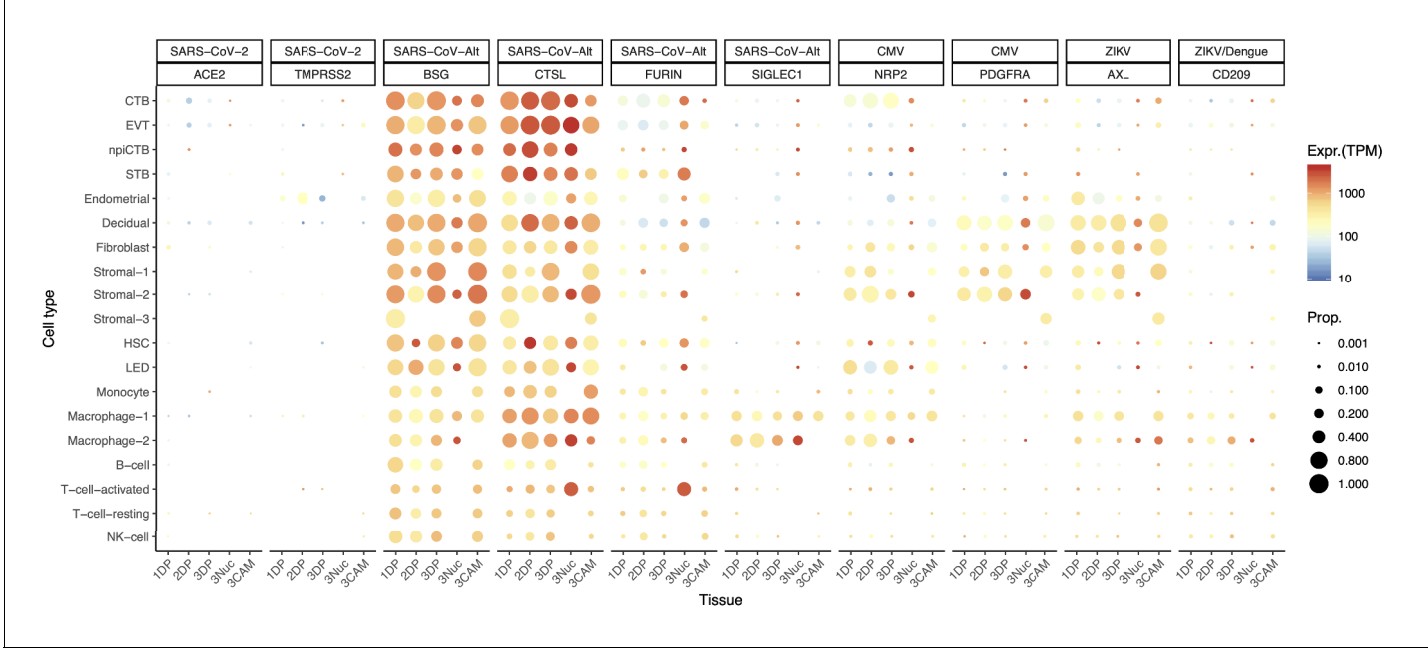

**Figure 2.** Dot plot depicting the expression of different viral receptors/molecules used by SARS-CoV-2, CMV, and ZIKV. Each row represents a different cell type, and columns are grouped first by virus type, receptor/molecule gene, and placental tissue/time-of sampling (1DP, 2DP and 3DP represent the first, second, and third trimester, 3Nuc represents the third trimester nuclei, and 3CAM represents the third trimester chorioamniotic membranes). The size of the dot represents the proportion of cells that express the receptor with more than zero transcripts, and the color represents the average gene expression for the subset of cells expressing that gene in transcripts per million (TPM). Cell type abbreviations used are: STB, Syncytiotrophoblast; EVT, Extravillous trophoblast; CTB, cytotrophoblast; HSC, hematopoietic stem cell; npiCTB, non-proliferative interstitial cytotrophoblast; LED, lymphoid endothelial decidual cell.

The online version of this article includes the following figure supplement(s) for figure 2:

**Figure supplement 1.** Gene expression values for BSG across tissues collected by the GTEx project.

**Figure supplement 2.** Dot plot depicting the expression of different viral receptors/molecules used by virus that caused congenital infection.

levels of priming by themselves (*Shirato et al., 2017*; *Shirato et al., 2018*; *Iwata-Yoshikawa et al., 2019*) when tested with SARS-CoV-1, yet this has not been verified for SARS-CoV-2. Given that the placental tissues are enriched in maternal and fetal macrophages (*Pique-Regi et al., 2019*), and that a subset of these immune cells expressing sialoadhesin (SIGLEC1, also known as CD169) can contribute to viral spread during SARS-CoV-2 infection (*Chen, 2020*; *Park, 2020*), we also investigated the expression of SIGLEC1 in this study. As expected, SIGLEC1 was expressed by macrophages in the placenta and chorioamniotic membranes and, to a lesser extent, in T cells (*Figure 2*, CoV-Alt). However, even if the virus could infect the placental/decidual macrophages expressing SIGLEC1, this is not sufficient for viral spreading. The expression of ADAM17 was also investigated in the placental tissues as this metalloproteinase competes with TMPRSS2 in ACE2 processing (*Heurich et al., 2014*). The placenta and chorioamniotic membranes highly expressed ADAM17 (*Figure 2—figure supplement 2*); however, only cleavage by TMPRSS2 results in augmented SARS-S-driven cell entry (*Heurich et al., 2014*). While these CoV-Alt molecules may be used for SARS-CoV-2 infection, they are likely to be less efficient than ACE2 and TMPRSS2, which are already targeted for antiviral interventions (*Hoffmann et al., 2020*); yet, new candidate host:viral interacting proteins and possible drugs are being investigated (*Gordon et al., 2020*).

Given that the main mediators for cell entry of SARS-CoV-2 were minimally expressed by the human placenta, we also investigated whether the receptors for congenital viruses such as CMV (*Stegmann and Carey, 2002*; *Arechavaleta-Velasco et al., 2002*; *Aronoff et al., 2017*; *Pereira et al., 2017*; *Al-Haddad et al., 2019*; *Faure Bardon et al., 2020*) and ZIKV (*Coyne and Lazear, 2016*; *Adams Waldorf et al., 2016*; *Aldo et al., 2016*; *Cao et al., 2017*; *Aagaard et al., 2017*; *Mysorekar, 2017*; *Valentine et al., 2018*; *Adams Waldorf et al., 2018*; *Dudley et al., 2018*; *Walker et al., 2019*; *Nelson et al., 2020*), which are known to infect and cross the placenta, were

detectable using our pipeline. Known receptors for CMV include NRP2 (*Martinez-Martin et al., 2018*), PDFGRA (*Martinez-Martin et al., 2018*), and CD46 (*Stein et al., 2019*). Notably, all of these receptors were highly expressed in several placental cell types (*Figure 2*, CMV and *Figure 2—figure supplement 2*). Next, we investigated the expression of the AXL receptor for ZIKV (*Richard et al., 2017*; *Persaud et al., 2018*) as well as other related molecules such as CD209 (*Carbaugh et al., 2019*) and TYRO3 (*Oliveira and Peron, 2019*). Consistent with vertical transmission, AXL, the preferred receptor for ZIKV, was highly expressed by the cells of the human placenta and chorioamniotic membranes throughout gestation (*Figure 2*, ZIKV). The expression of CD209 was mainly found in the maternal and fetal macrophage subsets, as expected (*Svensson et al., 2011*; *Swieboda et al., 2020*). Yet, the expression of TYRO3 was low (*Figure 2—figure supplement 2*), consistent with the view that TAM receptors are not essential for ZIKV infection (*Hastings et al., 2017*). The expression of other viral receptors involved in congenital disease was also documented in the placental tissues (*Figure 2—figure supplement 2*).

## Conclusion

In conclusion, the single-cell transcriptomic analysis presented herein provides evidence that SARS-CoV-2 is unlikely to infect the placenta and fetus since its canonical receptor and protease, ACE2 and TMPRSS2, are only minimally expressed by the human placenta throughout pregnancy. In addition, we showed that the SARS-CoV-2 receptors are not expressed by the chorioamniotic membranes in the third trimester. However, viral receptors utilized by CMV, ZIKV, and others are highly expressed by the human placental tissues. While transcript levels do not always correlate with protein expression, our data indicate a low likelihood of placental infection and vertical transmission of SARS-CoV-2. However, it is still possible that the expression of these proteins is much higher in individuals with pregnancy complications related to the renin-angiotensin-aldosterone system (RAAS), which can alter the expression of ACE2 (*Herse et al., 2007*; *Alexandre et al., 2020*). The cellular receptors and mechanisms that could be exploited by SARS-CoV-2 are still under investigation (*Gordon et al., 2020*), yet single-cell atlases can help to identify cell types with a similar transcriptional profile to those that are known to participate in COVID-19.

## Materials and methods

### Data availability

Placental and decidual scRNA-seq data from first-trimester samples were downloaded through ArrayExpress (E-MTAB-6701). Data for third-trimester samples previously collected by our group are available through NIH dbGAP (accession number phs001886.v2.p1), and newly generated second-trimester scRNA-seq and third-trimester snRNA-seq data are being deposited into the same repository (*Supplementary file 1*). All software and R packages used herein are detailed in the 'scRNA-seq and snRNA-seq data analysis.' Scripts detailing the analyses are also available at https://github.com/piquelab/sclabor (*Pique-Regi, 2020*; copy archived at https://github.com/elifesciences-publications/sclabor).

### Sample collection and processing, single-cell/nuclei preparation, library preparation, and sequencing
#### Human subjects

Placental tissues were obtained immediately after a clinically indicated delivery from (i) a patient diagnosed with placenta accreta at 18 weeks of gestation and (ii) 32 patients spanning different conditions in the third trimester (*Supplementary file 2*). A sample of the basal plate of the placenta including the decidua basalis and placental villi tissue was (i) dissociated as previously described (*Pique-Regi et al., 2019*) for scRNA-seq or (ii) preserved in RNAlater and subsequently frozen for snRNA-seq. The collection and use of human materials for research purposes were approved by the Institutional Review Boards of the Wayne State University School of Medicine and NICHD. All participating women provided written informed consent prior to sample collection.

## Single-cell preparation

Cells from the placental villi and basal plate were isolated by enzymatic digestion using previously described protocols with modifications (*Pique-Regi et al., 2019*; *Tsang et al., 2017*; *Xu et al., 2015*). Briefly, placental tissues were homogenized using a gentleMACS Dissociator (Miltenyi Biotec, San Diego, CA) either in an enzyme cocktail from the Umbilical Cord Dissociation Kit (Miltenyi Biotec) or in collagenase A (Sigma Aldrich, St. Louis, MO). After digestion, homogenized tissues were washed with ice-cold 1X phosphate-buffered saline (PBS) and filtered through a cell strainer (Fisher Scientific, Durham, NC). Cell suspensions were then collected and centrifuged at 300 x g for 5 min. at 4°C. Red blood cells were lysed using a lysing buffer (Life Technologies, Grand Island, NY). Next, the cells were washed with ice-cold 1X PBS and resuspended in 1X PBS for cell counting using an automatic cell counter (Cellometer Auto 2000; Nexcelom Bioscience, Lawrence, MA). Lastly, dead cells were removed from the cell suspensions using the Dead Cell Removal Kit (Miltenyi Biotec), and cells were counted again to determine final viable cell numbers.

## Single-cell library preparation using the 10x genomics platform

Viable cells were utilized for single-cell RNAseq library construction using the Chromium Controller and Chromium Single Cell 3' Version 3 Kit (10x Genomics, Pleasanton, CA), following the manufacturer's instructions. Briefly, viable cell suspensions were loaded into the Chromium Controller to generate gel beads in emulsion (GEM), with each GEM containing a single cell as well as barcoded oligonucleotides. Next, the GEMs were placed in the Veriti 96-well Thermal Cycler (Thermo Fisher Scientific, Wilmington, DE) and reverse transcription was performed in each GEM (GEM-RT). After the reaction, the complementary (c)DNA was cleaned by using Silane DynaBeads (Thermo Fisher Scientific) and the SPRIselect Reagent Kit (Beckman Coulter, Indianapolis, IN). Next, the cDNA was amplified using the Veriti 96-well Thermal Cycler and cleaned using the SPRIselect Reagent Kit. Indexed sequencing libraries were then constructed using the Chromium Single Cell 3' Version 3 Kit, following the manufacturer's instructions.

cDNA was fragmented, end-repaired, and A-tailed using the Chromium Single Cell 3' Version 3 Kit, following the manufacturer's instructions. Next, adaptor ligation was performed using the Chromium Single Cell 3' Version 3 Kit, followed by post-ligation clean-up using the SPRIselect Reagent Kit to obtain the final library constructs, which were then amplified using PCR. After performing a post-sample index double-sided size selection using the SPRIselect Reagent Kit, the quality and quantity of the DNA were analyzed using the Agilent Bioanalyzer High Sensitivity Chip (Agilent Technologies, Wilmington, DE). The Kapa DNA Quantification Kit for Illumina platforms (Kapa Biosystems, Wilmington, MA) was used to quantify the DNA libraries, following the manufacturer's instructions.

## Single-nuclei sample preparation

We developed a new protocol to isolate nuclei from frozen placenta samples, based on DroNc-seq (*Habib et al., 2017*) and an early version of the protocol developed by the Martelotto lab (https://www.protocols.io/view/frankenstein-protocol-for-nuclei-isolation-from-f-3eqgjdw). For each placenta sample, 1 mm frozen placenta biopsy punches were collected and immediately lysed with ice-cold lysis buffer (10 mM Tris-HCl, pH 7.5, 10 mM NaCl, 3 mM MgCl2, 2% BSA, 0.2 U/µl ROCHE Protector RNase Inhibitor, and 0.1% IGEPAL-630) for 5 min. During incubation the samples were gently mixed by swirling the tube twice and collected by centrifugation at 500 x g for 5 min at 4°C. The process was repeated twice for a total of 3 cycles of lysis (5 min long each). Next, the pellets were washed with ice-cold nuclei suspension buffer (1X PBS containing 2% BSA and 0.2 U/µl ROCHE Protector RNase Inhibitor) and filtered through a 30 µm cell strainer (Fisher Scientific). Nuclei suspensions were then collected and centrifuged at 500 x g for 5 min at 4°C. Nuclei were counted using a Countess II FL (Thermo Fisher Scientific, Durham, NC). All samples exhibited 100% cell death with DAPI staining, indicative of complete cell lysis. Nuclei were then utilized for single-nuclei RNAseq library construction using the Chromium Controller and Chromium Single Cell 3' version 2 kit (10x Genomics), following the manufacturer's instructions.

## Sequencing

Libraries were sequenced on the Illumina NextSeq 500 in the Luca/Pique-Regi laboratory and in the CMMG Genomics Services Center (GSC). The Illumina 75 Cycle Sequencing Kit was used with 58 cycles for R2, 26 for R1, and 8 for I1.

## scRNA-seq and snRNA-seq data analyses

Raw fastq files were downloaded from previously established resources (as detailed in 'Data Availability'), and the new sequencing data were processed using Cell Ranger version 3.0.0 from 10X Genomics for de-multiplexing. The fastq files were then aligned using kallisto (*Bray et al., 2016*), and bustools (*Melsted, 2019*) summarized the cell/gene transcript counts in a matrix for each sample, using the 'lamanno' workflow for scRNA-seq and the 'nucleus' workflow for snRNA-seq. Each sample was then processed using DIEM (*Alvarez, 2019*) to eliminate debris and empty droplets for both scRNA-seq and snRNA-seq. To avoid the loss of cells that may express viral receptors, we did not exclude cell doublets from the analyses included in this report, which should have negligible effects on the results and conclusions. All count data matrices were then normalized and combined using the 'NormalizeData,' 'FindVariableFeatures,' and 'ScaleData' methods implemented in the Seurat package in R (Seurat version 3.1, R version 3.6.1) (*Hafemeister and Satija, 2019*) and (*Stuart et al., 2019*). Afterward, the Seurat 'RunPCA' function was applied to obtain the first 50 principal components, and the different batches and locations were integrated and harmonized using the Harmony package in R (*Korsunsky et al., 2019*). The top 30 harmony components were then processed using the Seurat 'runUMAP' function to embed and visualize the cells in a two-dimensional map via the Uniform Manifold Approximation and Projection for Dimension Reduction (UMAP) algorithm (*McInnes et al., 2020*; *Becht et al., 2019*). To label the cells, the Seurat 'FindTransferAnchors' and 'TransferData' functions were used for each group of locations separately to assign a cell-type identity based on our previously labeled data as reference panel (as performed in *Pique-Regi et al., 2019*). Cell type abbreviations used are: STB, Syncytiotrophoblast; EVT, Extravillous trophoblast; CTB, cytotrophoblast; HSC, hematopoietic stem cell; npiCTB, non proliferative interstitial cytotrophoblast; LED, lymphoid endothelial decidual cell. Visualization of viral receptor gene expression was performed using the ggplot2 (*Wickham, 2011*) package in R with gene expression values scaled to transcripts per million (TPM) and to the proportion of cells expressing the transcript within a given cell type (*Booeshaghi and Pachter, 2020*).

## Bulk gene expression data analysis of ACE2 and TMPRSS2 in the placental tissues

Gene expression data for the study by *Kim et al., 2009* was available from the www.ebi.ac.uk/microarray-as/ae/ database (entry ID: E-TABM-577), while data for the study by *Toft et al., 2008* is available in our data repertoire. The mas5calls function from the *affy* package in Bioconductor was used to determine presence above background of each probeset corresponding to a given gene (*Gautier et al., 2004*).

## Acknowledgements

We thank the physicians, nurses, and research assistants from the Center for Advanced Obstetrical Care and Research, the Intrapartum Unit, and the PRB Clinical Laboratory for their help with collecting and processing samples.

## Additional information

### Funding

| Funder | Grant reference number | Author |
| --- | --- | --- |
| National Institutes of Health | HHSN275201300006C | Roberto Romero |
| Wayne State University | Perinatal Research Initiative | Adi L Tarca<br>Nardhy Gomez-Lopez |

The funders had no role in study design, data collection and interpretation, or the decision to submit the work for publication.

## Author contributions

Roger Pique-Regi, Conceptualization, Resources, Data curation, Formal analysis, Supervision, Investigation, Visualization, Methodology, Writing - original draft, Writing - review and editing; Roberto Romero, Conceptualization, Supervision, Funding acquisition, Investigation, Methodology, Writing - original draft, Project administration, Writing - review and editing; Adi L Tarca, Resources, Data curation, Formal analysis, Methodology, Writing - original draft, Writing - review and editing; Francesca Luca, Resources, Data curation, Supervision, Investigation, Methodology, Writing - original draft, Writing - review and editing; Yi Xu, Resources, Data curation, Formal analysis, Validation, Methodology, Writing - review and editing; Adnan Alazizi, Data curation, Validation, Methodology, Writing - review and editing; Yaozhu Leng, Resources, Data curation, Methodology, Writing - review and editing; Chaur-Dong Hsu, Resources, Project administration, Writing - review and editing; Nardhy Gomez-Lopez, Conceptualization, Data curation, Supervision, Investigation, Writing - original draft, Writing - review and editing

## Author ORCIDs

Roger Pique-Regi (iD) https://orcid.org/0000-0002-1262-2275
Roberto Romero (iD) https://orcid.org/0000-0002-4448-5121
Adi L Tarca (iD) http://orcid.org/0000-0003-1712-7588
Francesca Luca (iD) http://orcid.org/0000-0001-8252-9052
Nardhy Gomez-Lopez (iD) https://orcid.org/0000-0002-3406-5262

## Ethics

Human subjects: The collection and use of human materials for research purposes were approved by the Institutional Review Board of the Wayne State University School of Medicine and NICHD [IRB# 110605MP2F(RCR), IRB# 082403MP2F(5R), and IRB# 031318MP2F]. All participating women provided written informed consent prior to sample collection.

## Decision letter and Author response

Decision letter https://doi.org/10.7554/eLife.58716.sa1
Author response https://doi.org/10.7554/eLife.58716.sa2

---

# Additional files

## Supplementary files

• Supplementary file 1. Summary of all the single cell resources analyzed using existing and new data.

• Supplementary file 2. Clinical and demographic characteristics of the study population from which placental samples were collected for snRNAseq studies.

• Supplementary file 3. Bulk gene expression data analysis of ACE2 and TMPRSS2 in the placental tissues.

• Transparent reporting form

## Data availability

Placenta and decidua scRNA-seq data from first-trimester samples were downloaded through ArrayExpress (E-MTAB-6701). Data for third-trimester samples previously collected by our group are available through NIH dbGAP (accession number phs001886.v2.p1), and newly generated second-trimester scRNA-seq and third-trimester snRNA-seq data are deposited in the same repository.

The following dataset was generated:

| Author(s) | Year | Dataset title | Dataset URL | Database and Identifier |
|---|---|---|---|---|
| Pique-Regi R | 2020 | Single Cell Transcriptional Signatures of the Human Placenta | https://www.ncbi.nlm.nih.gov/projects/gap/cgi-bin/study.cgi?study_id=phs001886.v2.p1 | dbGaP, phs001886.v2.p1 |

The following previously published dataset was used:

| Author(s) | Year | Dataset title | Dataset URL | Database and Identifier |
|---|---|---|---|---|
| Vento-Tormo | 2018 | Reconstructing the human first trimester fetal-maternal interface using single cell transcriptomics - 10x data | https://www.ebi.ac.uk/arrayexpress/experiments/E-MTAB-6701/ | ArrayExpress, E-MTAB-6701 |

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
