## [Decision Letter]

**Acceptance summary:**

We believe this timely manuscript fills a critical gap in understanding vertical transmission biology of SARS-CoV-2 and other viruses. These results will help form the foundation for a broad network of single cell/nucleus profiles that can collectively help to dissect mechanisms relevant to the current pandemic. Thank you for the quick scientific progress from your team.

**Decision letter after peer review:**

Thank you for submitting your article "Does the human placenta express the canonical cell entry mediators for SARS-CoV-2?" for consideration by *eLife*. Your article has been reviewed by two peer reviewers, one of whom is a member of our Board of Reviewing Editors, and the evaluation has been overseen by Marianne Bronner as the Senior Editor. The reviewers have opted to remain anonymous.

The reviewers have discussed the reviews with one another and the Reviewing Editor has drafted this decision to help you prepare a revised submission.

Summary:

The paper by Pique-Regi and collaborators is of high interest in the current SARS-CoV-2 pandemics, trying to dissect molecular mechanisms behind the apparent lack of placental viral transfer to the fetus. In contrast to ZIka and cytomegalovirus infections in pregnancy, this new coronavirus does not seem to infect pregnant women more than others, nor do they get sicker if infected. Also, a vertical transmission to the fetus seems unlikely, but the molecular mechanisms for this fetal protection are unknown. Therefore, this paper fills an important knowledge gap by its findings of a lacking co-expression of two canonical receptor pathways for SARS-CoV-2 cellular infection (ACE2 and TMPRSS2). Overall, this is a timely and well-written manuscript that fills an important niche in the scientific literature. This is a strong work that only need a few revisions to increase clarity.

Essential revisions:

1) The authors claim that the new snRNA-seq data should help to identify multi-nucleated cells, like STB cells, which they do find in only (or predominantly) in the snRNA-seq data. However, in the UMAP plot in Figure 1A, these cells appear to be in two different clusters that mostly overlap what looks like fibroblasts (in purple) or immune cells (blue, light blue). Some explanation as to why the STB cells appear across these two otherwise distinct clusters would be helpful.

---

## [Author Response]

Essential revisions:1) The authors claim that the new snRNA-seq data should help to identify multi-nucleated cells, like STB cells, which they do find in only (or predominantly) in the snRNA-seq data. However, in the UMAP plot in Figure 1A, these cells appear to be in two different clusters that mostly overlap what looks like fibroblasts (in purple) or immune cells (blue, light blue). Some explanation as to why the STB cells appear across these two otherwise distinct clusters would be helpful.

We thank the reviewers for this comment. The snRNA-seq-derived STB clusters shown in Figure 1A (lime green) appear to be close yet do not completely overlap with the scRNA-seq-derived CTB (dark purple) and npi CTB (green). There is also some but incomplete overlap with NK-cell (dark blue) and T-cell-activated (forest green) clusters, among others, probably as a result of the higher background/soup RNA coming from these cell types. We have added a supplementary figure (Figure 1—figure supplement 1) that displays the high expression of STB specific genes (GH2, CSH1, and CGA) in the STB cluster. We respectfully believe that further investigation into the nature of these two separate clusters is beyond the scope of this article, as there is no co-expression of ACE2 and TMPRSS2 in either of them.